# GhMAX2 Contributes to Auxin-Mediated Fiber Elongation in Cotton (*Gossypium hirsutum*)

**DOI:** 10.3390/plants13152041

**Published:** 2024-07-25

**Authors:** Zailong Tian, Haijin Qin, Baojun Chen, Zhaoe Pan, Yinhua Jia, Xiongming Du, Shoupu He

**Affiliations:** 1National Nanfan Research Institute (Sanya), Chinese Academy of Agricultural Sciences, Sanya 572024, China; zltian2021@163.com (Z.T.); duxiongming@caas.cn (X.D.); 2Zhengzhou Research Base, National Key Laboratory of Cotton Bio-Breeding and Integrated Utilization, School of Agricultural Sciences, Zhengzhou University, Zhengzhou 450001, China; chenbaojun@caas.cn (B.C.); jiayinhua@caas.cn (Y.J.); 3State Key Laboratory of Cotton Bio-breeding and Integrated Utilization, Institute of Cotton Research, Chinese Academy of Agricultural Sciences, Anyang 455099, China; qinhaijin2023@163.com (H.Q.); panzhaoe@caas.cn (Z.P.)

**Keywords:** crosstalk, degradation, GhIAA17, strigolactones

## Abstract

Strigolactones (SLs) represent a new group of phytohormones that play a pivotal role in the regulation of plant shoot branching and the development of adventitious roots. In cotton (*Gossypium hirsutum*, *Gh*), SLs play a crucial role in the regulation of fiber cell elongation and secondary cell wall thickness. However, the underlying molecular mechanisms of SL signaling involved in fiber cell development are largely unknown. In this study, we report two SL-signaling genes, *GhMAX2-3* and *GhMAX2-6*, which positively regulate cotton fiber elongation. Further protein—protein interaction and degradation assays showed that the repressor of the auxin cascade GhIAA17 serves as a substrate for the F-box E3 ligase GhMAX2. The in vivo ubiquitination assay suggested that GhMAX2-3 and GhMAX2-6 ubiquitinate GhIAA17 and coordinately degrade GhIAA17 with GhTIR1. The findings of this investigation offer valuable insights into the roles of GhMAX2-mediated SL signaling in cotton and establish a solid foundation for future endeavors aimed at optimizing cotton plant cultivation.

## 1. Introduction

Upland cotton (*Gossypium hirsutum*) is the most fruitful cultivated cotton, accounting for approximately ~97% of the global cotton fiber yield [1]. Cotton fiber is derived from the seed coat, consisting of fuzz and lint fibers, with lint fibers representing highly specialized and elongated single cells. This unique cellular structure makes lint fibers an excellent model for investigating the regulatory mechanisms underlying cell polar elongation. The developmental process of cotton fiber cells encompasses five distinct stages: cell initiation, elongation, transitional wall thickening, cell wall thickening, and maturation [2]. Notably, the trait of fiber length largely determines fiber quality.

Strigolactones (SLs) are derived from carotenoids and were initially discovered in the cotton root exudates [3]. Subsequent studies have demonstrated that SLs also function in stimulating the branching of hyphae in arbuscular mycorrhizal fungi [4]. SLs have diverse functions in regulating plant development and response to stress [5,6,7,8,9,10]. In rice, the pyramiding of the beneficial alleles involved in SL biosynthesis has contributed to the new ‘Green Revolution.’ The precursor for SL biosynthesis is derived from all-trans β-carotene through the action of an isomerase DWARF27 (D27) and subsequent cleavage by the CAROTENOID CLEAVAGE DIOXYGENASES 7 and 8 (CCD7 and CCD8) to produce 9-cis-β-apo-10-carotene (CL) [11,12,13,14]. In *Arabidopsis*, the *MORE AXILLARY GROWTH* (*MAX*) family genes, including *MAX1*, *MAX2*, *MAX3*, and *MAX4*, participate in SL biosynthesis, perception, and signaling [15,16]. The bioactive SLs are perceived by α/β hydrolase DWARF14 (D14), and D14 interacts with the F-box protein MAX2 (D3 in rice, *Oryza sativa*) to form the Skp-Cullin-F-box (SCF) E3 ubiquitin ligase complex D14-SCFMAX2, leading to the degradation of the SL repressors SUPPRESSOR OF MAX2-LIKE 6 (SMXL6), 7, and 8 (D53 in rice), which, in turn, release the repression of downstream target genes and trigger SL responses [17,18,19].

SLs fulfill diverse crucial functions in regulating plant growth, development, and response to abiotic and biotic stresses. In *Arabidopsis*, SLs inhibit lateral shoot branching [5], promote primary root growth, and suppress lateral root development under normal growth conditions [20]. In rice, SLs inhibit the tiller number and mesocotyl elongation in darkness [6,21]. Moreover, SLs play a pivotal role in plant responses to abiotic stress, such as drought, salt, cold, and heat stress [7,22,23]. Recent studies have unveiled the involvement of SLs in promoting cotton fiber elongation and secondary cell wall thickness [24,25,26]. However, the detailed molecular mechanisms in whichSLs modulate cotton fiber development remain elusive.

In this study, we find that the core components of SL signaling, *GhMAX2-3* (*Gh_A12G105500*) and *GhMAX2-6* (*Gh D12G102700*), positively regulate cotton fiber elongation. GhMAX2-3 and GhMAX2-6 interact with GhIAA17 (*Gh_D10G103900*), a repressor of auxin signaling, to mediate the ubiquitination and degradation of GhIAA17. Therefore, GhIAA17 acts a molecular bridge to connect SL- and auxin-signaling pathways.

## 2. Results

### 2.1. GhMAX2-3 and GhMAX2-6 Regulate Cotton Fiber Elongation

The previous study has shown that MAX2 encodes a subunit of an SCF E3 ligase and acts as a key positive regulator of the SL-signaling pathway [16]. Bioinformatic and RT-qPCR analysis revealed that upland cotton has six orthologs of *Arabidopsis MAX2*, in which *GhMAX2-3* and *GhMAX2-6* are highly expressed during the fiber elongation stage (around 5- to 15-day post-anthesis, 5–15 DPA) in J02-508 (Appendix A) (PRJNA634606). Thus, we explored the biological functions of GhMAX2-3 and GhMAX2-6 in cotton fiber development. We used a virus-induced, gene-silencing (VIGS) system to knock down the expression of *GhMAX2-3* and *GhMAX2-6* (Figure 1A,B). Phenotypic analysis revealed that *GhMAX2-3-* and *GhMAX2-6*-silencing plants produced shorter mature fibers compared with the WT and vector control (Figure 1C,D). These results indicated that GhMAX2-3 and GhMAX2-6 positively regulate fiber elongation.

### 2.2. GhMAX2-3 and GhMAX2-6 Interact with GhIAA17

To investigate the underlying mechanism of GhMAX2-mediated fiber elongation, we conducted a yeast two-hybrid (Y2H) assay using GhMAX2-3-BD as a bait. Following screening against a cDNA library from cotton fibers, a total of six poteintial GhMAX2-3 interacting candidates were identified (Appendix A), including GhIAA17. Y2H assays confirmed that both GhMAX2-3 and GhMAX2-6 interact with GhIAA17 (Figure 2A). We further verified the GhMAX2-3-6–GhIAA17 interactions in tobacco leaves using firefly luciferase complementation imaging (LCI) (Figure 2B,C). To further confirm the interactions in vivo, we performed co-immunoprecipitation (Co-IP) assays by transiently co-expressing GhIAA17-Flag and Myc-GhMAX2-3 or GhMAX2-6 fusion proteins in tobacco leaves. As expected, we found that Myc-GhMAX2-3 and Myc-GhMAX2-6 could be co-immunoprecipitated with GhIAA17-Flag (Figure 2D).

### 2.3. GhMAX2-3 and GhMAX2-6 Mediate the Ubiquitination and Degradation of GhIAA17

It has been established that auxin facilitates the formation of its coreceptor complex involving F-box proteins TIR1/AFBs and transcription regulators AUX/IAA proteins (AUX/IAAs), triggering to the ubiquitination and sbusequent degradation of AUX/IAAs, thus abolishing the inhibitory effects of AUX/IAAs on ARF transcription factors, to activate auxin-responsive gene expression [27,28]. We found that GhIAA17 was rapidly degraded upon treatment with IAA in a dose dependent (Appendix A). Furthermore, we observed that the 26S proteasome inhibitor MG132 completely suppressed the IAA-triggered GhIAA17 degradation (Appendix A), indicating that GhIAA17 is targeted for degradation via the 26S proteasome pathway. Given that GhMAX2 encodes a E3 ligase, we speculated that GhMAX2-3 and GhMAX2-6 might affect the accumulation of GhIAA17 protein. To test this, we first carried out an in vivo ubiquitination assay in cotton protoplasts. The constructs encoding Flag-Ub, Myc-GhIAA17, GhMAX2-3/6-GFP, or GFP control were transiently expressed in cotton protoplasts. Total proteins were extracted and immunoprecipitated with protein A/G agarose beads, coupled with anti-Flag antibody, followed be subjected to western blotting analysis using an anti-Myc antibody. We detected higher levels of polyubiquitinated GhIAA17 protein in protoplasts co-expressing GhMAX2-3/6-GFP and Myc-GhIAA17 compared with control (Figure 3A). We then evaluated whether GhMAX2-3/6 would modulate GhIAA17 degradation in a degradation assay. Myc-GhIAA17, along with GhMAX2-3/6-GFP or GFP control, were co-expressed in cotton protoplasts for 16 h and then treated with 200 μM CHX for indicated times. Total proteins were extracted and followed by immunoblot detection with an anti-Myc antibody. The results showed that GhMAX2-3 and GhMAX2-6 promoted the degradation of GhIAA17 (Figure 3B). These findings suggest that GhMAX2-3 and GhMAX2-6 promote GhIAA17 degradation in an auxin-independent manner.

### 2.4. GhTIR1 and GhMAX2 Have Additive Effects on the Degradation of GhIAA17

Both GhTIR1 and GhMAX2 exert F-box proteins, which mediate the auxin and SL-induced substrate degradation, raising the possibility that GhMAX2 and GhTIR1 might synergistically regulate GhIAA17 degradation. The Y2H and LCI assays showed that GhTIR1 could interact with GhIAA17 (Figure 4A,B). To elucidate the components or signaling pathways implicated in auxin/SL-mediated cotton fiber development, we conducted in vivo degradation assays. We found that the GhIAA17 protein was degraded faster when GhTIR1 and GhMAX2-3 or GhMAX2-6 were co-expressed than when GhTIR1 was expressed alone (Figure 4B). Our results suggest that GhMAX2 and GhTIR1 collaboratively regulate GhIAA17 degradation.

### 2.5. GhIAA17 Inhibits Cotton Fiber Elongation

To determine the potintial role of GhIAA17 in fiber development, the expression patterns of *GhAUX/IAA* family members were investigated in different stages of fiber development. We found that *GhIAA17* was preferentially expressed in rapid-elongation fibers (Appendix A). To characterize the biological function of GhIAA17, we constructed a *GhIAA17*-silencing vector and performed a VIGS assay. The mature fibers from *GhIAA17*-silenced plants were longer than those from the control plants (Figure 5). These results suggest that GhIAA17 negatively regulates fiber elongation.

## 3. Discussion

Strigolactones (SLs) are a class of phytohormones that regulate various aspects of plant development, such as shoot branching and the tiller number [5,20,22]. Although considerable research has been conducted on these functions and the interactions between SL signaling other plant hormone pathways, our understanding of SL power at the cellular level remains limited. Upland cotton (*Gossypium hirsutum*), a commercially significant crop, serves as the primary source of natural fiber. The precise roles and underlying mechanisms through which SLs modulate fiber development in cotton have not yet been fully elucidated. In this study, we show that GhMAX2 plays an important role during the fiber elongation stage. Silencing *GhMAX2-3* and *GhMAX2-6* in cotton resulted in shorter mature fibers (Figure 1). A previous report has demonstrated that the content of *epi*-5DS (a canonical endogenous SL) increases gradually during the fiber elongation stage (around 5–20 DPA), and the treatment of the cotton ovule with rac-GR24 promotes fiber elongation [24], implying that SLs play an essential role in fiber development. SLs trigger the interaction of the D14 receptor with the SMXL repressor and then induce the formation of a D14-SMXL-SCF^MAX2^ complex for the ubiquitination and degradation of SMXL6, 7, and 8 through the 26S proteasome [28,29]. A recent study has demonstrated that GhMAX2 contributes to cotton fiber elongation [26]. However, knowledge about the molecular mechanisms of GhMAX2 promoting fiber elongation is scarce. In this study, we identified a new target of GhMAX2 through Y2H screening. Further investigation demonstrated that GhMAX2-3 and GhMAX2-6 could induce GhIAA17 degradation (Figure 3). However, whether the GhMAX2-mediated GhIAA17 degradation is SL-dependent remains to be explored. Furthermore, we cannot exclude the possibility that there are other proteins downstream of GhMAX2-3 and GhMAX2-6, such as GhSMXLs, which merits further detailed dissection.

The phytohormone auxin also plays an essential role in regulating various aspects of plant development, including vascular differentiation, cell elongation, and division [30,31]. The stability of Aux/IAAs plays a decisive role in the regulation of auxin-signaling pathways as transcription repressors. In cotton, which AUX/IAAs control the single cell elongation remains unclear. In this study, we found that auxin triggers the degradation of GhIAA17 and verifies its function in cotton fiber growth. Silencing *GhIAA17* in cotton reduced the length of the fiber cells. We have not identified the GhARFs upstream of GhIAA17, which transcriptionally regulate fiber development. The accumulation of the plant hormone IAA in cotton fiber could substantially increase the number and length of lint fibers [32]. However, the mechanism by which the AUX/IAA–GhARF pair regulates fiber development remains largely unclear. Fifty-six ARF genes were identified in upland cotton, and *GhARF2* and *GhARF18* subfamily genes may play important roles in regulating cotton seed fiber initiation [33]. A recent study has characterized the functions of two GhIAA proteins (GhAXR2 and GhSHY2) in cotton fiber cell development and identified more downstream GhARFs (GhARF7-1, GhARF19-1, GhARF17-5, and GhARF18-6) that positively regulate fiber cell elongation [34]. The potential GhARFs interacting with GhIAA17 need to be further clarified in the future.

The interplay and interdependency of phytohormones, commonly referred to as phytohormone crosstalk, are prevalent mechanisms governing plant development at the single-cell level [35]. In cotton, cytokinin disrupts auxin accumulation through the modulation of the auxin efflux carrier PIN-FORMED 3a in the ovule epidermis to inhibit fiber initiation, implicating the involvement of cytokinin-auxin crosstalk in the regulation of fiber cell initiation [36]. Previous studies have provided evidence for crosstalk between the SL and auxin pathways. For instance, SL acts as a negative regulator of shoot branching by impeding the basipetal auxin transport [37,38,39]. Recent research has demonstrated that auxin induces the expression of the SL biosynthesis genes MAX3 and MAX4 of the *Arabidopsis* stem in an AXR1-dependent manner [40]. These findings point to a feedback mechanism underlying the crosstalk between SL and auxin signaling. The relationship between SL- and auxin-signaling components during cotton fiber elongation was unexplored, though the individual contributions of SL and auxin to plant development have been examined previously. In our study, we reveal that the components of the SL and auxin pathways have additive effects on GhIAA17 degradation, thus releasing upstream GhARFs (Figure 6). In summary, our data reveal a SL–auxin crosstalk-mediated regulatory mechanism for promoting fiber elongation in cotton.

## 4. Materials and Methods

### 4.1. Plant Materials and Methods

Upland cotton (*Gossypium hirsutum* cv. TM-1) seeds were surface-sterilized with 70% (*v*/*v*) ethanol for 5 min and 0.1% (*m*/*v*) HgCl_2_ for 15 min and then washed with sterile water. The sterilized seeds were germinated on half-strength MS (1/2 MS) medium under a 16:8 h, light/dark cycle at 28 °C for 7 days. The hypocotyls from these seedlings were used for protoplasts preparation. The cotton seeds were germinated in soil and cultivated in controlled greenhouse conditions, with 28 °C day/22 °C night, 40–60% relative humidity and a 16 h light/8 h dark cycle. These seedlings were used for VIGS experiments.

### 4.2. RNA Extraction and RT-qPCR

The total RNA was extracted from 10 DPA cotton fibers using the FastPure Universal Plant Total RNA Isolation Kit (Vazyme Biotech Co., Ltd., Nanjing, China), and 1 μg of total RNA was used for first-strand cDNA synthesis with the Superscript™ First-Strand Synthesis System (Invitrogen, Carlsbad, CA, USA). Transcript abundance was quantified by RT-qPCR assays using SYBR green master mix (Toyobo, Japan). The reactions were performed using the Roche Light Cycle 480 II instrument (Roche, Basel, 560 Switzerland) programmed as follows: an initial denaturation at 95 °C for 3 min, followed by 40 cycles of 95 °C for 25 s, 56 °C for 30 s, and 72 °C for 30 s. The melting curve was generated from 65 °C to 95 °C. Fluorescence signals were automatically acquired at the end of each cycle. The relative expression levels of the corresponding genes were calculated using the 2^−ΔΔCT^ method [41]. Three independent biological replicates were performed for each gene.

### 4.3. Primers

The primers used in this study are listed in Appendix A.

### 4.4. Y2H Assay

For the Y2H assay, the coding sequences of *GhMAX2-3* and *GhMAX2-6* were amplified and inserted into a pGBKT7 vector (Clontech, Takara, Japan) as baits. The coding sequence of *GhIAA17* was cloned and ligated into a pGADT7 vector as a prey. The bait vector and prey vector were co-transformed into the yeast strain Y2HGold. Following growth on SD [DDO for -Trp/-Leu] medium, the transformants were dropped on selective plates [QDO for -Trp/-Leu/-His/-Ade] 30 °C for 5 days. AD-T/BD-53 was used as the positive control, and AD-T/BD-Lam was used as the negative control [42].

### 4.5. LCI Assay

The full-length CDS of *GhMAX2-3*, *GhMAX2-6*, and *GhIAA17* was cloned into the pCAMBIA1300-cLUC or pCAMBIA1300-nLUC binary vectors to generate N-terminus (nLUC)- or C-terminus (cLUC)-fused and truncated luciferase tags on the target proteins, respectively. All constructs were paired and transformed into *Agrobacterium* strain GV3101 (pSoup P19). After being resuspended in buffer (0.1 mM acetosyringone, 10 mM MgCl2, 10 mM MES [pH 5.6]), the cultures were syringe-injected into tobacco leaves. The infiltrated leaves were incubated in the dark for 12 h, followed by 48 h in the light at 25 °C. The infiltrated leaves were sprayed with 1 mM D-luciferin potassium salt solution (CL6930, Coolaber, Beijing, China), and the LUC activity was examined using the Tanon 4600 Plant Imaging System.

### 4.6. Co-IP Assay

The GhIAA17-Flag, Myc-GhMAX2-3, and Myc-GhMAX2-6 vectors were transfected into *Agrobacterium* strain GV3101. The *Agrobacterium* cells containing individual constructs were co-infiltrated into tobacco leaves. The soluble proteins were extracted using a native extraction buffer 1 containing 50 mM Tris-MES, pH 8.0, 0.5 M sucrose, 1 mM MgCl_2_, 10 mM EDTA (ethylenediaminetetraacetic acid), 5 mM dithiothreitol (DTT), 50 μM MG132, and 1 × protease inhibitor cocktail (MCE^®^) [43]. After centrifugation at 14,000× *g* at 4 °C for 15 min, the supernatant was collected. The Flag-fused GhIAA17 was immunoprecipitated using BeyoMag™ Anti-Flag Beads (Beyotime) at 4 °C for 1 h. Then the beads were washed three times with 1× phosphate-buffered saline (PBS) containing 150 mM NaCl. The co-immunoprecipitated samples were boiled with loading buffer and run on 10% SDS-PAGE for western blotting with anti-Myc (1:2000; Abcam) and anti-Flag (1:5000; Abcam) antibodies.

### 4.7. In Vivo Ubiquitination Assay

In vivo ubiquitination assays were performed as described previously with some modifications [44]. The coding sequences of *GhIAA17* and *GhMAX2-3/6* were cloned into the pM999 vector, tagged with Myc and Flag to generate Myc-GhIAA17, GhMAX2-3-GFP, and GhMAX2-6-GFP vectors. These constructs, together with Flag-Ub, were co-transfected into cotton protoplasts and incubated for 16 h in the presence of 5 μM MG132. The total proteins were extracted using ubiquitination buffer containing 50 mM Tris-HCl [pH 7.5], 150 mM NaCl, 1% Triton X-100, 1% sodium deoxycholate, 0.1% SDS, 1×protease inhibitor cocktail (MCE^®^), 1 mM PMSF, and 100 μM MG132, and subjected to immunoprecipitation with an anti-Flag antibody, coupled with protein A/G agarose beads at 4 °C for 3 h. The beads were then washed three times with PBS containing 150 mM NaCl. An equal amount of immunoprecipitated products was separated by 10% SDS-PAGE and detected with anti-Myc antibody, which enables only polyubiquitinated Myc-GhIAA17 to be detected.

### 4.8. In Vivo Degradation Assay

For the GhMAX2-mediated GhIAA17 degradation, the GhIAA17-Flag and GhMAX2-3-GFP or GhMAX2-6-GFP vectors (6 μg) were co-transfected into cotton protoplasts. For the transient expression assay in the cotton protoplasts, the embryogenic callus was incubated in the digestion solution 1.5% Cellulase R10 (Yakult Pharmaceutical Industry), 0.4% Macerozyme R10 (Yakult Pharmaceutical Industry, Tokyo, Japan), 1% Hemi Cellulase (Sigma, St. Louis, MO, USA), 0.4 M Mannitol, 20 mM KCl, 20 mM MES (pH 5.7), 10 mM CaCl_2_, and 0.1% BSA. The vectors were transformed in protoplasts using PEG 4000 (Sigma). The transformed protoplasts were cultured at 25 °C in the dark for 16 h. Then, the protoplasts were treated with 200 μM CHX for the indicated times [45]. Total proteins were extracted using a protein extraction buffer (20 mM Tris-HCl [pH 7.5], 150 mM NaCl, 1% Triton X-100, 1×protease inhibitor cocktail (MCE^®^), and 1 mM PMSF. The Myc-GhSLR1 protein level was detected by performing western blotting analysis using anti-Myc and anti-Actin antibodies.

### 4.9. Virus Induced Gene Silencing (VIGS)

An improved virus-induced, gene-silencing (VIGS) method [46] was used to silence the expression of *GhMAX2-3*, *GhMAX2-6*, and *GhIAA17* in cotton. The −300 bp fragments of *GhMAX2-3*, *GhMAX2-6*, and *GhIAA17* were amplified by PCR and inserted into a pCLCrV vector. Plasmids of pCLCrVB, pCLCrVA, pCLCrV-GhMAX2-3, pCLCrV-GhMAX2-6, and pCLCrV-GhIAA17 were individually introduced into *Agrobacterium* strain LBA4404. *Agrobacterium* cultures containing pCLCrVB, pCLCrVA, pCLCrV-GhMAX2-3, pCLCrV-GhMAX2-6, and pCLCrV-GhIAA17 were cultured in 2–3 mL YEP medium at 28 °C for 12–16 h, then transferred to 50 mL YEP and cultured for another 12–16 h. The *Agrobacterium* were collected and resuspended to OD600 = 0.8 with infiltration buffer (10 mM MES [pH 5.4], 10 mM MgCl_2_, and 200 mM acetosyringone [AS]) and set in the dark for 3 h. LBA4404 cells containing pCLCrVB were mixed with cells harboring pCLCrV-GhMAX2-3, pCLCrV-GhMAX2-6, and pCLCrV-GhIAA17 and infiltrated into the true leaves of cotton seedlings. Three biological replicates were performed in each experiment, and more than 20 plants were measured in each repetition.

### 4.10. Statistical Analysis

The quantification of the relative protein levels in the immunoblot assay was performed using ImageJ with Java 8 software. All statistical analyses were conducted in GraphPad Prism 10. Statistical significance among the data was determined by Tukey’s multiple comparison tests.

## 5. Conclusions

In this study, two SL-signaling genes, *GhMAX2-3* and *GhMAX2-6*, positively regulate cotton fiber elongation. The repressor of auxin cascade GhIAA17 serves as a substrate for the F-box E3 ligase GhMAX2. GhMAX2-3 and GhMAX2-6 ubiquitinate GhIAA17 and coordinately degrade GhIAA17 with GhTIR1. We reveal that the components of the SL and auxin pathways have additive effects on GhIAA17 degradation, thus releasing upstream GhARFs. In summary, our data reveal a SL–auxin crosstalk-mediated regulatory mechanism for promoting fiber elongation in cotton.

## Figures and Tables

**Figure 1 plants-13-02041-f001:**
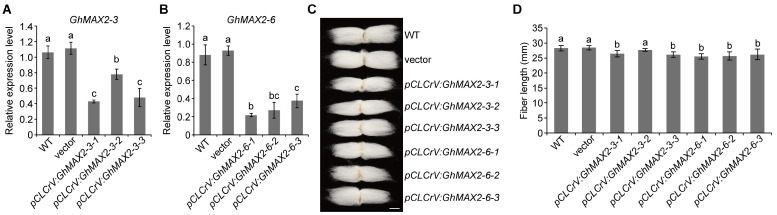
GhMAX2-3 and GhMAX2-6 regulate fiber length. (**A**,**B**) Relative expression level of *GhMAX2-3* (**A**) and *GhAMX2-6* (**B**) in 10 DPA fibers from wild-type and VIGS cotton plants. The empty vector was used as the negative control. *GhUBQ7* was used as the internal control. Data are represented as the mean ± SD (n = 3). (**C**) Image of mature fibers from WT, *GhMAX2-3-*, and *GhMAX2-6*-silencing plants. Scale bar = 1 cm. (**D**) Analysis of fiber length shown in (**C**). Data are represented as the mean ± SD (n > 10). Different letters in (**A**,**B**,**D**) at the top of each column indicate a significant difference at *p* < 0.05 determined by Tukey’s HSD test.

**Figure 2 plants-13-02041-f002:**
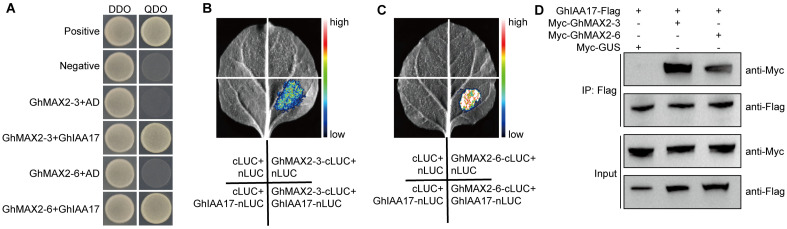
GhMAX2-3 and GhMAX2-6 interact with GhIAA17. (**A**) Y2H assay showing the interactions of GhMAX2-3 and GhMAX2-6 with GhIAA17. DDO represents SD/-Leu/-Trp medium, and QDO represents SD/-Ade/-His/-Leu/-Trp medium. AD-T/BD-53 was used as the positive control, and AD-T/BD-Lam was used as the negative control. (**B**,**C**) LCI assays showing that GhMAX2-3 and GhMAX2-6 interact with GhIAA17 in tobacco leaves. GhIAA17 and GhMAX2-3/6 were fused with N- and C-terminal of luciferase protein, respectively. The high–low scales represent LUC activities. (**D**) Co-IP assay showing the interactions of GhMAX2-3/6 with GhIAA17 in vivo. GhIAA17-Flag and Myc-GhMAX2-3/6 or Myc-GUS (β-glucuronidase) constructs were transiently expressed in tobacco leaves.

**Figure 3 plants-13-02041-f003:**
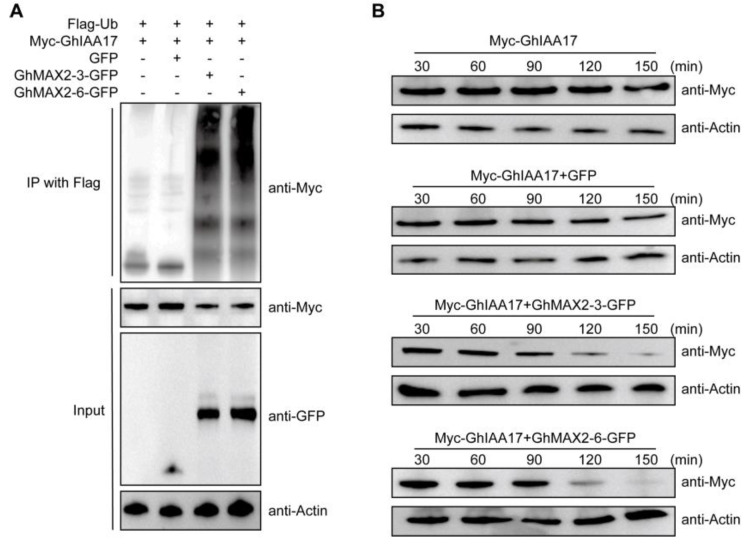
GhMAX2-3 and GhMAX2-6 mediate the polyubiquitination and degradation of GhIAA17. (**A**) Ubiquitination of GhIAA17 by GhMAX2-3 and GhMAX2-6 in cotton protoplasts. Constructs encoding GhMAX2-3-GFP or GhMAX2-6-GFP, Myc-GhIAA17, and Flag-tagged Ub (Flag-Ub) proteins were co-transfected into cotton protoplasts in the presence of 5 μM MG132 for 16 h at 25 °C. Total proteins were extracted and incubated with anti-Flag beads, and the polyubiquitinated GhIAA17 protein was detected by immunoblotting with an anti-Myc antibody. (**B**) In vivo degradation assay showing the effect of GhMAX2-3 and GhMAX2-6 on GhIAA17 degradation in cotton protoplasts. These constructs were co-transfected into cotton protoplasts. After being cultured for 16 h, the protoplasts were treated with 200 μM CHX for the indicated times. Total proteins were extracted and subjected to western blotting analysis using anti-Myc and anti-Actin antibodies. Actin was used as the internal control.

**Figure 4 plants-13-02041-f004:**
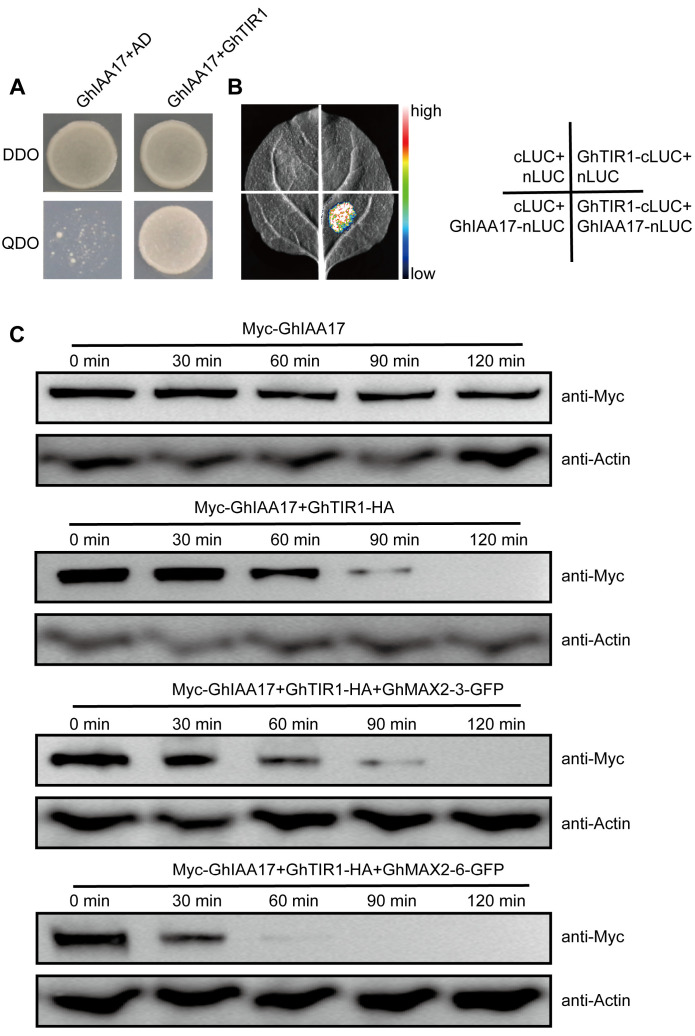
GhMAX2-3 and GhAMX2-6 promote GhTIR1-induced GhIAA17 degradation. (**A**) Y2H assay showing the interaction of GhTIR1 with GhIAA17. (**B**) LCI assay showing the GhTIR1-GhIAA17 interaction in vivo. (**C**) In vivo degradation assays in cotton protoplasts. Myc-GhIAA17 and GhTIR1-HA, with or without GhMAX2-3/6 constructs, were co-transfected into cotton protoplasts and allowed to be expressed for 16 h at 25 °C. The protoplasts were treated with 200 μM CHX, and the samples were collected at the indicated time points. Myc-GhIAA17 abundance was detected by immunoblotting with an anti-Myc antibody. Actin was used as the internal control.

**Figure 5 plants-13-02041-f005:**
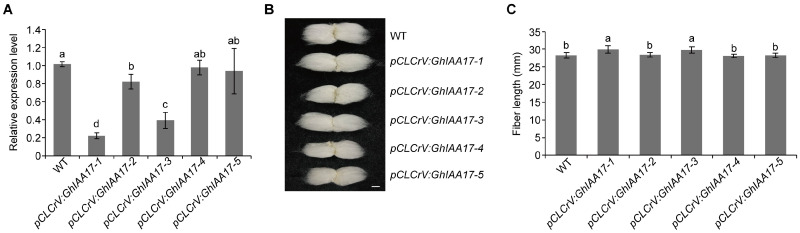
Silencing of *GhIAA17* increases fiber length. (**A**) Relative expression level of *GhIAA17* in 10 DPA fibers from wild-type and VIGS cotton plants. The empty vector was used as the negative control. *GhUBQ7* was used as the internal control. Data are represented as the mean ± SD (n = 3). (**B**) Image of mature fibers from WT and *GhIAA17*-silencing plants. Scale bar = 1 cm. (**C**) Analysis of fiber length shown in (**B**). Data are represented as the mean ± SD (n > 10). Different letters in (**A**,**C**) at the top of each column indicate a significant difference at *p* < 0.05 determined by Tukey’s HSD test.

**Figure 6 plants-13-02041-f006:**
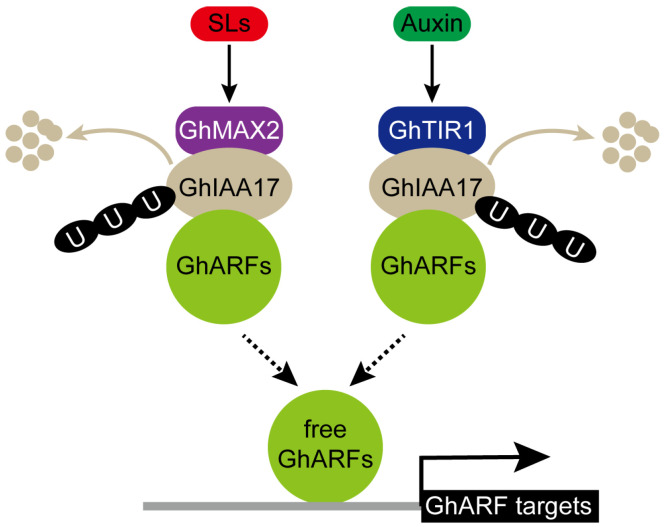
A proposed model illustrating that GhMAX2 modulating auxin signaling via binding to GhIAA17 in cotton fibers. GhIAA17 could respond to both SL and auxin signaling. SL and auxin trigger GhMAX2- and GhTIR1-mediated GhIAA17 degradation. Then, the released GhARFs bind to the promoters of their targets to regulate cotton fiber elongation.

## Data Availability

Data are contained within the article and Appendix A.

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
