# Peer review of "GhMAX2 Contributes to Auxin-Mediated Fiber Elongation in Cotton (Gossypium hirsutum)"

_plants, 2024, doi:10.3390/plants13152041_

Round 1

Reviewer 1 Report

Comments and Suggestions for Authors

Dear Authors,

I have reviewed your manuscript with the ID plants-3086306, titled "GhMAX2 contributes to auxin-mediated cotton fiber elongation" I appreciate the timely subject of the paper. I have a few suggestions to improve your manuscript:

Keywords: "cotton fiber" and "GhMAX2" should be changed because they already exist in the title. Please write keywords in alphabetical order.

Abstract: Please write some basic information about the experiment.

Materials and Methods:

Please present the statistical analysis used.

Why did you opt for in vitro germination of cotton seeds even though you transferred them to soil after 7 days? If this germination method is important, you should justify it in the discussion.

This paper is important for research in the field and should be considered for publication after addressing the issues mentioned above.

Thank you for your contribution and the efforts you have put into this study.

Author Response

Comments 1: I have reviewed your manuscript with the ID plants-3086306, titled "GhMAX2 contributes to auxin-mediated cotton fiber elongation" I appreciate the timely subject of the paper. I have a few suggestions to improve your manuscript:

Keywords: "cotton fiber" and "GhMAX2" should be changed because they already exist in the title. Please write keywords in alphabetical order.

Response 1: Thanks for your suggestion. We have replaced ‘cotton fiber’ and ‘GhMAX2’ with ‘crosstalk’ and ‘degradation’. Line 26.

Comments 2: Abstract: Please write some basic information about the experiment.

Response 2: Thanks for your suggestion. We have recomposed some sentences and included the basic experiment information in the ‘Abstract’ section. Line 19-21.

Comments 3: Materials and Methods: Please present the statistical analysis used.

Response 3: Thanks for your meticulous suggestion. We have provided the statistical analysis in the ‘Materials and Methods’ section. Line 363-367.

Comments 4: Why did you opt for in vitro germination of cotton seeds even though you transferred them to soil after 7 days? If this germination method is important, you should justify it in the discussion.

Response 4: We apologize for any confusion caused. In fact, the sterile seedlings grown on 1/2 MS were used for protoplasts transfection. For the VIGS assay, the seeds were germinated in soil. We have clarified this point in the revised manuscript as follows: “The hypocotyls from these seedlings were used for protoplasts preparation. The cotton seeds were germinated in soil and cultivated in controlled greenhouse conditions, with 28 °C day/22 °C night, 40 %-60 % relative humidity and a 16-h light/8-h dark cycle. These seedlings were used for VIGS experiments.” Line 268-272.

Reviewer 2 Report

Comments and Suggestions for Authors

I checked your manuscript and described comments below.

This paper provides an excellent analysis of the Strigolactones (SLs) signal gene, MAX2.

I would recommend that you consider the following points.

1.     The abstract does not state that Gh is cotton (Gossypium hirsutum, Gh). I think it would be better to state that.

2.     I think the title should also be "cotton (Gossypium hirsutum)".

3.     If possible, I recommend that you include the following papers in references.

Richard J Challis, Jo Hepworth, Céline Mouchel, Richard Waites, Ottoline Leyser, A role for more axillary growth1 (MAX1) in evolutionary diversity in strigolactone signaling upstream of MAX2, Plant Physiol. 2013 Apr;161(4):1885-902. doi: 10.1104/pp.112.211383.

4.     I think it would be better to show the relationships between the genes in the Discussion section in a diagram, as this would make it easier to understand the results of this paper.

I don't think this paper has new various major mistakes or grammatical problems.

Author Response

Comments 1: The abstract does not state that Gh is cotton (Gossypium hirsutum, Gh). I think it would be better to state that.

Response 1: Thank for your professional suggestion. We have inserted the ‘Gossypium hirsutum, Gh’ in Abstract section. Line 16.

Comments 2: I think the title should also be "cotton (Gossypium hirsutum)".

Response 2: We have revised the title as follows: “GhMAX2 contributes to auxin-mediated fiber elongation in cotton (Gossypium hirsutum)”. Thank you very much.

Commends 3: If possible, I recommend that you include the following papers in references. Richard J Challis, Jo Hepworth, Céline Mouchel, Richard Waites, Ottoline Leyser, A role for more axillary growth1 (MAX1) in evolutionary diversity in strigolactone signaling upstream of MAX2, Plant Physiol. 2013 Apr;161(4):1885-902. doi: 10.1104/pp.112.211383.

Response 3: Agreed. We have included the reference in our revised version. Line 418-419.

Comments 4: I think it would be better to show the relationships between the genes in the Discussion section in a diagram, as this would make it easier to understand the results of this paper.

Response 4: Very professional suggestion. We have provided a new Figure 6 to illustrate the coordinated effect of SL and auxin on GhIAA17.

Reviewer 3 Report

Comments and Suggestions for Authors

The study by Tian et al. offers valuable insights into the role of strigolactone (SL) signaling in cotton fiber development, highlighting GhMAX2-3 and GhMAX2-6 as key regulators through the ubiquitination and degradation of GhIAA17. To strengthen the conclusion that GhMAX2-3 and GhMAX2-6 are highly expressed during the fiber elongation stage, I suggest including qPCR verification experiments. Additionally, in the discussion section, compare the results of the previously published article (He et al., Journal of Integrative Agriculture, 2022) with the current study, discussing the similarities and focusing on the new findings. It is also recommended to include a model diagram in it.

1. In Lines 67-68, the previous publication should be properly cited.

2. The authors used bioinformatic analysis to reveal that GhMAX2-3 and GhMAX2-6 are highly expressed during the fiber elongation stage and then focused on the detailed gene function study. Have the authors used qPCR to verify if these genes are highly expressed in the cultivar used in this study?

3. All gene names should be italicized, and protein names should be displayed in regular font. Please ensure uniformity throughout the manuscript.

4. In Line 92, for Table S1, it is suggested that the authors provide clear gene annotations.

5. In Figures 2B and 2C, a heatmap bar is needed to indicate the extent of interaction. For Figure 2D, when the authors only infiltrate with GhIAA17, why can the MYC fusion protein be detected using anti-Myc? Please check this carefully.

6. In Lines 180-181, the authors should mark which gene is GhIAA17 in Fig. S3 and verify if it is preferentially expressed in elongated fibers using qPCR.

7. In the discussion section, suggest the authors draw a model to illustrate how GhTIR1 and GhMAX2 have additive effects on the degradation of GhIAA17, making it easier for readers to follow GhMAX2’s role in modulating fiber development.

8. To ensure that similar experiments can be more easily replicated by others, it is suggested that the authors provide more detailed protocols. All protocols, including the LCI assay, Co-IP assay, in vivo ubiquitination assay, and in vivo degradation assay, should be described in detail rather than simply citing previously published papers.

Author Response

Comments: The study by Tian et al. offers valuable insights into the role of strigolactone (SL) signaling in cotton fiber development, highlighting GhMAX2-3 and GhMAX2-6 as key regulators through the ubiquitination and degradation of GhIAA17. To strengthen the conclusion that GhMAX2-3 and GhMAX2-6 are highly expressed during the fiber elongation stage, I suggest including qPCR verification experiments. Additionally, in the discussion section, compare the results of the previously published article (He et al., Journal of Integrative Agriculture, 2022) with the current study, discussing the similarities and focusing on the new findings. It is also recommended to include a model diagram in it.

Response: Thank you very much for your professional comments. According to your suggestion, we have conducted RT-qPCR assay to confirm that GhMAX2-3 and GhMAX2-6 are highly expressed in elongated cotton fibers (Figure S1B). Furthermore, we have inserted some sentences in the Discuss section to clarify where are consistencies and discrepancies to previous work: “A recent study has demonstrated that GhMAX2 contributes to cotton fiber elongation [26]. However, knowledge on molecular mechanisms of GhMAX2 promoting fiber elongation is scarce. In this study, we identified a new target of GhMAX2 through Y2H screening.” Line 214-217. Moreover, a proposed working model was presented in Figure 6 to demonstrate the relationship between GhMAX2 and GhIAA17.

Comments 1: In Lines 67-68, the previous publication should be properly cited.

Response 1: Sorry for the careless. We have cited the related reference in our revised manuscript. Line 71.

Comments 2: The authors used bioinformatic analysis to reveal that GhMAX2-3 and GhMAX2-6 are highly expressed during the fiber elongation stage and then focused on the detailed gene function study. Have the authors used qPCR to verify if these genes are highly expressed in the cultivar used in this study?

Response 2: Thanks for your suggestion. Now we have performed RT-qPCR assay to detect the relative expression level of GhMAX2. The result showed that GhMAX2-3 and GhMAX2-6 are highly expressed in 10 DPA fibers, while the expression of GhMAX2-1, GhMAX2-2, GhMAX2-4, and GhMAX2-5 are undetectable (new Figure S1B). This result is consistent with the transcriptome data (Figure S1A).

Comments 3: All gene names should be italicized, and protein names should be displayed in regular font. Please ensure uniformity throughout the manuscript.

Response 3: Thanks for your correction. We have checked throughout the manuscript and revised the improper font format.

Comments 4: In Line 92, for Table S1, it is suggested that the authors provide clear gene annotations.

Response 4: Thanks for your suggestion. We have revised Table S1 and provide detailed information regarding gene annotations.

Comments 5: In Figures 2B and 2C, a heatmap bar is needed to indicate the extent of interaction. For Figure 2D, when the authors only infiltrate with GhIAA17, why can the MYC fusion protein be detected using anti-Myc? Please check this carefully.

Response 5: Thank you very much for pointing this out. We have provided the high-low bars for Figure 2B-C. Actually, in Co-IP assay, we used GUS gene as a negative control, which has the similar protein size with GhMAX2-3/6. We have revised Figure 2D in the revised version.

Comments 6: In Lines 180-181, the authors should mark which gene is GhIAA17 in Fig. S3 and verify if it is preferentially expressed in elongated fibers using qPCR.

Response 6: Thanks for your suggestion. We have denoted Gh_D10G103900 (IAA17) with red in Figure S13. Further RT-qPCR analysis showed that GhIAA17 was highly expressed in elongated fibers (new Figure S4).

Comments 7: In the discussion section, suggest the authors draw a model to illustrate how GhTIR1 and GhMAX2 have additive effects on the degradation of GhIAA17, making it easier for readers to follow GhMAX2’s role in modulating fiber development.

Response 7: Agreed a lot. Now we have provided a new Figure 6 to illustrate the additive effects of GhMAX2 and GhTIR1 on GhIAA17 degradation.

Comments 8: To ensure that similar experiments can be more easily replicated by others, it is suggested that the authors provide more detailed protocols. All protocols, including the LCI assay, Co-IP assay, in vivo ubiquitination assay, and in vivo degradation assay, should be described in detail rather than simply citing previously published papers.

Response 8: Thanks for your suggestion. We have provided the detailed information in the ‘Materials and Methods’ section as you suggested.